# Analysis of Structural Characteristics and Psychometric Properties of the SarQoL^®^ Questionnaire in Different Languages: A Systematic Review

**DOI:** 10.3390/ijerph19084561

**Published:** 2022-04-10

**Authors:** María Visitación Martínez-Fernández, Irene Sandoval-Hernández, Alejandro Galán-Mercant, Manuel Gonzalez-Sanchez, Jesús Martínez-Cal, Guadalupe Molina-Torres

**Affiliations:** 1Rehabilitation Service, Mutua Montañesa Hospital, 39012 Santander, Spain; mv.martinez@mutuamontanesa.es; 2Department of Physical Therapy, Faculty of Health Sciences, University of Granada-Campus of Melilla, C/Santander, 1, 52005 Melilla, Spain; isandoval@ugr.es; 3MOVE-IT Research Group, Department of Physical Education, Faculty of Education Sciences, University of Cádiz, 11519 Puerto Real, Spain; 4Department of Nursing and Physiotherapy, University of Cádiz, 11009 Cadiz, Spain; 5Biomedical Research and Innovation Institute of Cádiz (INIBICA) Research Unit, Puerta del Mar University Hospital, University of Cádiz, 11009 Cadiz, Spain; 6Institute of Biomedicine of Málaga (IBIMA), 29010 Malaga, Spain; mgsa23@uma.es; 7Department of Physiotherapy, Faculty of Health Sciences, University of Málaga, 29071 Malaga, Spain; 8Department of Nursing, Physiotherapy and Medicine, Faculty of Health Sciences, University of Almería, 04120 Almeria, Spain; jesus.martinez@ual.es (J.M.-C.); guada.lupe@ual.es (G.M.-T.)

**Keywords:** sarcopenia, quality of life, validation, questionnaire, cross-cultural adaptation, older adult, aging

## Abstract

Background: Sarcopenia is the gradual and global loss of muscle and its functions. Primary sarcopenia is associated with the typical changes of advanced aging and affects approximately 5–10% of the population. The Sarcopenia and Quality of Life (SarQoL^®^) questionnaire is composed of 55 items, 22 questions, and is organized into seven domains of quality of life. The main objective of this systematic review was to analyze the structural characteristics and psychometric properties of it, as well as to classify its measurement properties, its methodological quality, and the criteria as good measurement properties of the adaptations and validations made on the SarQoL^®^ questionnaire in different languages. Methods: A systematic review was carried out in the PUBMED, Web of Science, Cinahl, LatinIndex, and SCOPUS databases. The keywords used were: “SarQoL”, “assessment”, “sarcopenia”, “geriatric”, “PROM”, “quality of life”, and “questionnaire”, using the Boolean operator “AND”. All articles published up to 15 January 2022 were considered. Methodological quality and psychometric properties were assessed based on the COSMIN guidelines and the guidelines and general recommendations of PRISMA. Documents published in languages other than English were excluded, as well as versions of the SarQoL^®^ published in the form abstracts for conferences when the full text was not available. Results: A total of 133 articles were identified, 14 of which were included. The evaluated questionnaires and the structural characteristics and psychometric properties of each of them were collected. Conclusion: The different cross-cultural versions of the questionnaire showed good basic structural and psychometric characteristics for the evaluation of patients with sarcopenia.

## 1. Introduction

Sarcopenia is the gradual and global loss of the musculoskeletal system associated with low muscle quality and quantity, the most representative characteristic of which being muscle insufficiency linked to the loss of strength and dysfunction presented by patients suffering from this disorder [1,2]. Sarcopenia is currently associated with heterogeneous causality, including physiological, genetic, and environmental factors [3], and two different types are established according to their trigger: primary sarcopenia that is caused by age-related tissue deterioration as the only apparent cause. The prevalence of SarQoL^®^ is over 5%, rising above 10% in patients over 70 years of age. It is detected and assessed between the ages of 50 and 70 with an annual progression of 0.5–2% [4,5]; secondary sarcopenia, where more than one cause can be established, is generally associated with previous diseases, such as an advanced organ failure, a systemic disease, an endocrine disease, an inflammatory disease, and those diseases related to nutrition including but not limited to malnutrition, nutrient malabsorption, gastrointestinal disorders, anorexia, and aphagia [1,6].

Taking into account the physical consequences of this pathology and the relevant increase among the aging population, sarcopenia is considered a significant public health problem with a great economic and social intervention [6] because of the decrease in muscle mass and quality implies several fatal consequences, such as physical disability, risk of falls, loss of mobility, and depression that may well be reinforced, leading to a poor quality of life, dependency, hospitalization, and mortality [3,7,8,9]. The effects of this process are reflected into the health-related quality of life (HRQL) defined by the World Health Organization (WHO) as the perception of individuals, their position in life, culture and values, and the association with their goals, concerns, perspectives, and standards [9]. Research on the quality of life in patients with sarcopenia has been studied on numerous occasions over the years, firstly by means of generic questionnaires, such as the SF-36 or the EQ-5D, but precisely because of their generic nature, they are not sensitive to sarcopenia except for very specific domains [10,11]. Subsequently and at present, the SarQoL^®^ questionnaire is used, specifically developed to assess quality of life in patients with sarcopenia, with the ability to discriminate between sarcopenic and non-sarcopenic subjects and those individuals with muscle dysfunction. This self-administered questionnaire collects psychological, social, well-being, and physical effects that assess variation in the quality of life of the sarcopenic population over time [9,12]. It is composed of 55 items and 22 questions, structured in seven domains such as: physical and mental health, locomotion, body composition, functionality, activities of daily living, leisure activities, and fears [13,14]. SarQoL^®^ was created in 2015 by Beaudart et al. [14] in French, and has been translated into 31 languages with 34 versions available and five in progress [15]. Their psychometric properties have been developed in 13 of the translated versions, and all of them have followed a translation and validation protocol developed by their authors to guarantee the homogeneity of the process [9].

That is why the main objective of this systematic review was to analyze the structural characteristics and psychometric properties of it, as well as classifying its measurement properties, its methodological quality, and the criteria as good measurement properties of the cross-cultural adaptations and validations made on the SarQoL^®^ questionnaire in different languages.

## 2. Materials and Methods

### 2.1. Protocol

A systematic review of the literature was carried out, which was registered in the PROSPERO database (CRD No.: 42022303755) and the guidelines and general recommendations of the PRISMA declaration [16] and COSMIN guidelines [17,18] were followed.

### 2.2. Sources and Search

For developing this systematic review, a systematic search was carried out in the PUBMED, Web of Science, Cinahl, LatinIndex, and SCOPUS databases. The keywords used were: “SarQoL”, “assessment”, “sarcopenia”, “geriatric”, “PROM”, “quality of life”, and “questionnaire”. These terms were used in combination with the Boolean operators “AND” and “OR”. Articles published up to 15 January 2022, were considered.

### 2.3. Selection Criteria

The following selection criteria were considered for this review:

Inclusion criteria: Studies including cross-cultural adaptation and validation of the SarQoL^®^ questionnaire for assessing the quality of life in people with sarcopenia in any language other than the original language. Exclusion criteria: Studies having completed the adaptation phase but not having completed the validation phase, and studies that have been featured in the form of abstracts at conferences but have not been published in full text in any database, which made it impossible to analyze their psychometric properties.

### 2.4. Selection of Documents

The identified documents were submitted to the Rayyan platform (rayyan.qcri.org) [19] to collect, review, and evaluate citation titles and abstracts. First, the articles found as duplicated were eliminated, comprising a total of 64 documents. Subsequently, two researchers (G.M.-T., J.M.-C.) carried out an independent and blinded review and screening based on titles and abstracts, and those articles not meeting the inclusion criteria were eliminated. Conversely, those complying with the said criteria were selected and located for full-text reading. Furthermore, those articles arising doubts or in which the title and abstract did not reveal sufficient information to determine their inclusion or exclusion were also retrieved. Discrepancies were solved by a third reviewer (A.G.-M.).

### 2.5. Instrument

The original version of the SarQoL^®^ questionnaire was developed by Beaudart et al. in 2015 [14] and validated by the same authors two years later [8]. The questionnaire consists of 55 items translated into 22 questions, which in turn are organized into seven different dysfunction domains: physical and mental health, locomotion, body composition, functionality, activities of daily living, leisure activities, and fears [14]. The SarQoL^®^ questionnaire response options are a combination of Likert scales (3, 4, or 5 levels) and questions with different multiple-choice options. Although the questionnaire is easy to complete in just 10 min, there is also a short version of only 14 items, which the authors recommend where the original version may be too burdensome for respondents [20].

### 2.6. Results Synthesis and Data Extraction

All the articles finally selected were analyzed in order to identify the cross-cultural adaptation and collect information on the process of construction and validation of these tools. Likewise, the structural characteristics extracted from each cross-cultural adaptation were title, authors, year of publication, acronym, population, BMI, setting, diagnosis of sarcopenia, number of subjects with sarcopenia, number of subjects in the pilot phase, and number of subjects per item in the validation phase. On the other hand, the results of the extracted psychometric properties were test-retest reliability, internal consistency, and construct validity.

## 3. Result

After having identified 133 documents, 132 through the search in the databases and one [21] through the search in the reference lists of the selected articles, 68 were eliminated due to being duplicates. Of the 65 selected documents, 31 were finally excluded by title and abstract. The remaining 34 articles were subject to full-text examination, after which 21 were excluded, three because they only consisted of abstracts of a conference, and one because the validation phase had not been carried out. Finally, we had a total of 14 articles left for carrying out this systematic review; this entire selection process is shown in the flowchart below (Figure 1).

After reading the titles and applying the selection criteria to all the documents, a total of 14 cross-cultural adaptations were selected [5,12,13,21,22,23,24,25,26,27,28,29,30,31] in different languages, such as English, Romanian, Dutch, Polish, Greek, Lithuanian, Russian, Spanish, Ukrainian, Korean, Serbian, Chinese, and Turkish. The Hungarian version was not considered as the validation phase of the questionnaire had not been carried out. At the same time, the Persian, Czech, and Latvian versions were not included as their text was not fully available, and only an abstract of which was found in the World Congress on Osteoporosis, Osteoarthritis, and Musculoskeletal Diseases (WCO-IOF-ESCEO 2020): Poster abstracts [32].

Table 1 details the structural characteristics of the questionnaires: acronym, population, Body Mass Index (BMI), setting, diagnosis of sarcopenia/number of subjects with sarcopenia, number of subjects included in the piloting phase, and number of subjects included per item in the validation phase.

Furthermore, the number of questions and items were the same in all the cross-cultural adaptations of the SarQoL^®^ questionnaire included in this review, that is, all the adaptations into the different languages included 22 questions and 55 items. At the same time, the self-administration of time was only specified in two of the adaptations, in the English version, which was established in 10 min [30], and in the Spanish version in 10–15 min [5]. In addition, the diagnostic criteria for sarcopenia that were taken into account in each of the adaptations were those collected until 2018 by the European Working Group on Sarcopenia in Older People (EWGSOP); from the adaptations made in 2019 onwards, the diagnostic criteria included by the European Working Group on Sarcopenia in Older People, revised in early 2018 (EWGSOP2) were those taken into account, although only in the Chinese version of the SarQoL^®^ questionnaire other different diagnostic criteria for sarcopenia were taken into account, i.e., the Asian Working Group for Sarcopenia criteria (AWGS).

The psychometric properties of the questionnaires are shown in Table 2. They include reliability, internal consistency, and construct validity measured through convergent and divergent validity.

### 3.1. Content Validity

To evaluate the validity of the content in the 13 versions of the SarQoL^®^, the three criteria considered in the COSMIN guidelines were taken into account [33], including relevance, comprehensiveness, and comprehensibility. Twelve out of the thirteen studies analyzed the validity of the content, and six of them [22,23,24,26,27,29] considered the comprehensibility criterion. The validity of the content could not be evaluated because these aspects were doubtful or unclear, and so were considered as inconsistent. The patients were not asked about any of these three aspects, and as the authors evaluated the relevance and comprehensiveness and these were not shown clearly or in sufficient detail, so they were classified as inconsistent in relation to the criteria for measurement properties.

### 3.2. Structural Validity

None of the studies evaluated the structural validity of the SarQoL^®^, so the extent to which the scores obtained reflect an adequate dimensionality of the quality of life in patients with sarcopenia could not be analyzed. The COSMIN guidelines [17] recommend this property to be evaluated prior to internal consistency or cross-cultural validity.

### 3.3. Internal Consistency

The internal consistency was calculated for the 13 adapted versions of the SarQoL^®^ questionnaire using Cronbach’s alpha; they scored an internal consistency considered excellent α > 0.8 in all questionnaires, proving a high internal consistency. The range of Cronbach’s alpha values was between 0.866 of the Korean version [26] and 0.96 in the Greek version [23]. Thus, the classification of the criteria for a good measurement property was considered for all the studies as “indeterminate”, since the criteria were met because the structural validity according to the COSMIN guidelines had not been taken into account [17].

### 3.4. Test-Retest Reliability

The intraclass correlation coefficient (ICC) was used to test the reliability between the first and the second questionnaire according to the scores of the individual and general domains of the SarQoL^®^. An ICC greater than 0.7 is considered an acceptable reliability [34]. It was measured in all the adaptations included in this review, except in the Romanian [31] and Serbian [27] versions of the SarQoL^®^. ICC values ranged from 0.935 for the Russian version [24] to 0.99 of the Spanish and Polish versions [5,13]. The highest score was recorded for Domain D6 (Leisure activities) at 1.00 in the Polish version [12] and the lowest for Domain D7 (Fears) at 0.64, CI 95% (0.52–0.70) in the Greek version [23]. The Korean version did not specify the time elapsed between the first and second questionnaire and for the rest of the studies there was an interval of two weeks. Test-retest reliability was rated as a “sufficient” measurement property in the studies in which it was included.

### 3.5. Measurement Errors

Only one adaptation considered the measurement error referring to the Standard Error of Measurement (SEM). In the Greek version [23], SEM was reported on each of the SarQoL^®^ dimensions (D1: 2.42; D2: 3.15; D3: 6.95; D4: 2.7; D5: 5.04; D6: 6.23; D7: 9.17) and the total SEM score attained 2.75. The SEM is a parameter commonly used to indicate the amount of measurement error in an instrument, the interpretation of the measurement around a mean value, or the range within which the “true” value lies [35]. Other error measurements, such as the Minimal Important Change (MIC) or Limits of Agreement (LoA), were not considered in this study. Therefore, based on the criteria for a good measurement property, according to the COSMIN guidelines [17], the classification would be described as “indeterminate” [23] because the MIC has not been reported.

### 3.6. Construct Validity (Convergent Validity)

Convergent validity was reported in all studies [5,12,13,22,23,24,25,26,27,28,29,30,31]. In addition, all the studies established the correlations with the SF-36 and EQ-5D. Regarding the SF-36, although not applicable to all the studies included, they were correlated with the domains of physical functioning, role limitation due to physical problems, bodily pain, and general health and vitality. In the case of the Polish version, the SF-36v2^®^ PCS (physical component summary) and the SF-36v2^®^ MCS (mental component summary) were correlated, and, at the same time, the Ukrainian version was also correlated with the SF-36v2^®^ PCS. On the other hand, the EQ-5D was also included for the dimensions of utility score, mobility, and usual activities; self-care was also added in the Turkish version. In addition, the EQ-VAS was used in the Dutch, Polish, Spanish, and Ukrainian versions (see Table 2). Therefore, according to the COSMIN guidelines [17], based on the criteria for a good measurement property, the classification scored was considered as “insufficient” in three of the studies [5,23,26] because the results are not in agreement with the hypothesis that 75% of the correlations are ≥0.50. The rest were considered “sufficient”, as the results were in agreement with the 75% hypothesis (see Table 3).

### 3.7. Construct Validity (Divergent Validity)

Divergent validity was reported in all studies except for the Polish version [12]; the correlations were made based on the SF-36, the EQ-5D, and the HADS. The dimensions of the SF-36 included social functioning, role limitation due to emotional problems, mental health, role limitation due to physical problems (only in the Greek and Korean versions), emotional wellbeing (only in the Korean version), bodily pain (only in the Turkish version), and the Ukrainian version is the only one that included the SF-36 MCS dimension in divergent validity. Moreover, only in the Spanish version HADS anxiety and HADS depression were correlated. On the other hand, in relation to the correlation of the EQ-5D, the dimensions self-care, pain/discomfort, and anxiety/depression were included. Therefore, according to the COSMIN guidelines [17], and based on the criteria for a good measurement property, all the studies were considered “insufficient” in terms of classification of good measurement property, except the Spanish [5] and the Korean [26] versions, which were considered “sufficient”, since 75% of their indices were less than 0.30, and finally, the Polish version [12] was considered as “indeterminate” because it did not include divergent validity (see Table 3).

### 3.8. Criterion of Validity and Responsiveness

For the validity of the criterion, the COSMIN guidelines take into account the evaluation of this property based on the agreement with the hypothesis, using an external instrument or “gold standard” [17]. The measure considered by the authors of the COSMIN guidelines is the AUC (Area Under the Curve), which takes into account those values greater or lower than 0.70. It should be noted that no information on this property was reported; therefore, it could not be qualified. Similarly, responsiveness provides us with information to detect changes over time and considers the same AUC measurement using values of 0.70. None of the 13 validation studies of SarQoL^®^ evaluated this feature, nor did the original version take it into account [8]. That is why it did not qualify either.

### 3.9. Floor-Ceiling Effect

The ceiling or floor effect refers to the percentage of patients who obtained the highest scores (ceiling) or the lowest (floor), with percentages greater than 15% being considered significant [34,36]. They were analyzed in all the studies, although no floor effect was observed in any of them and only one study provided data on the ceiling effect, i.e., the Ukrainian version [25] in Domain 7 (fears) in which a ceiling effect of 28.6% could be found in 14 people with 100 points. However, no reference was made to the floor-ceiling effect in the Korean version [26]. The ceiling-floor effect is not considered a measurement property according to the COSMIN guidelines, although it has been considered in the referred studies.

### 3.10. Discriminative Power

In all versions, the discriminative power was considered since it is an instrument specifically designed to be used in sarcopenic populations; that is why the capacity of the questionnaire to differentiate between subjects with sarcopenia and those without sarcopenia must be taken into account. Furthermore, it is evaluated by comparing the total score of the SarQoL^®^ questionnaire with the scores of the individual domains [14]. In all studies, quality of life is better in subjects without sarcopenia compared with subjects diagnosed with sarcopenia. The discriminative power could not be evaluated, as it was not considered a measurement property by the COSMIN guidelines.

### 3.11. Methodological Quality

The “inadequate” methodological quality of most of the studies was due to the low number of samples equal to five times the number of elements, and only in three adaptations was the methodological quality very good [26,27,30] according to the COSMIN guidelines [18] (see Table 3). The methodological quality was evaluated based on the criteria of: (1) very good; seven subjects per item in samples ≥100 participants; (2) adequate; five subjects per item in samples ≥100 participants, or six subjects per item in samples <100 participants; (3) doubtful; five subjects per item in samples <100 participants; (4) inadequate; <5 subjects per item.

## 4. Discussion

The objective of this systematic review was to carry out an analysis of the validated questionnaires of the SarQoL^®^ with cross-cultural adaptation into different languages, for the evaluation of the quality of life in patients with sarcopenia, and to collect the structural characteristics, psychometric properties, as well as the classification of their measurement properties, their methodological quality, and the criteria as good measurement properties of all the versions of the questionnaires and, subsequently, to compare for identifying the most relevant ones to be used in clinical practice, as well as in the field of research. A total of 14 studies were identified and included in the analysis of this systematic review.

The adaptation studies included samples ranging from 10 to 25 subjects, except in the Spanish version [5] in which this was not reported, while in the original version [8] a total of 43 subjects and 12 experts were included. According to Beaton et al. [37], the ideal number of subjects for the pilot phase should be between 30–40 subjects, as recommended in the AAOS (American Academy of Orthopedic Surgeons) guide. Therefore, larger samples of subjects in the cross-cultural adaptation phase should be considered for future adaptations of the SarQoL^®^ in other languages.

In the validation phase, the sample in the different adaptations ranged between 49 and 699 subjects, while in the original version a total of 296 subjects were included. Only three adaptations [26,27,30] to other languages had a very good methodological quality, while the methodological quality of the original version [8] is adequate. Larger subject samples should be considered for future validation in other languages.

Structural validity was not considered in any of the SarQoL^®^ adaptations, and it was not considered in the original version either. Therefore, for future adaptations of this questionnaire, the analysis of its internal structure should be included, since it is important to know it in order to decide how the items should be combined within a scale or subscale [17]. Regarding internal consistency, all the adaptations showed an excellent Cronbach’s alpha greater than 0.7, and the original version refers to a Cronbach’s alpha of 0.87; each of the seven SarQoL^®^ domains ranged from 0.84 in D1-Mental Health to 0.89 in D6-Leisure Activities. Therefore, despite presenting an excellent Cronbach’s alpha, it would be convenient in future adaptations to analyze the structural validity to complete the internal validity.

To test the reliability of the SarQoL^®^, the test-retest was used in week two in all the studies except the Romanian [31] and the Serbian [27] versions. Likewise, a time span of two weeks elapsed in the original version [8]. On the other hand, the ICC was excellent in all versions of SarQoL^®^ and in their corresponding dimensions, as in the original version [8], except in the Dutch version [22] (domain 6 and 7) and in the Greek version [23] (domain 7) that scored a low test-retest reliability.

The measurement error was only included in the Greek version [23] and it was not considered in the original version of the SarQoL^®^. Therefore, in future adaptations of SarQoL^®^ this should be taken into account as a measurement property [17,34].

To analyze the validity of the construct, the correlations with SF-36 and EQ-5D were included for both convergent validity and divergent validity, except in the Spanish version [5], which also included the HADS for divergent validity. Regarding the original version [8], in addition to the SF-36 and EQ-5D, it was also correlated with the Mini Mental State Examination and the Mobility-Tiredness Scale. Regarding convergent validity, all versions showed a good correlation, except three of them [5,23,26] that presented an insufficient correlations; the original version shows a good convergent validity. In relation to the divergent validity, only two versions showed a good correlation [5,26], and in the original version [8] a good correlation is also recorded.

Furthermore, criterion validity and responsiveness were not reported in any of the SarQoL^®^ versions, nor in the original version. Therefore, in future adaptations and validations of this questionnaire into other languages, these measurement properties could be included to consider the changes recorded if some type of treatment is included.

In both the SarQoL^®^ versions and the original, no ceiling-floor effects were observed, except in the Ukrainian version [25] in which the ceiling effect only occurred in one of its domains. Moreover, in all versions of the SarQoL^®^ the discriminative power was taken into account in relation to the total score and by domains, in the same way as in the original version [8], with the exception that in the original version a logistic regression was used for comparing both groups (sarcopenia and non sarcopenia) as in the Chinese version [28].

### Strengths and Limitations

Although this is the first study that analyzes the psychometric and structural characteristics of a reference questionnaire for the evaluation of patients with sarcopenia, such as the SarQoL^®^, there are some limitations that must be indicated when interpreting the results reached. For example, although the search was carried out in five databases of worldwide relevance, there could be some versions of SarQoL^®^ not being collected in the aforementioned databases and, therefore, not included in this study. In addition, it is important to highlight that there are some limits within the studies themselves that should be corrected in future studies, since the evaluation of some psychometric properties could not be made. This was the case of structural validity. Therefore, in the future it would be convenient to design studies analyzing this psychometric property, which is highly significant for questionnaires validation.

Although the different versions of the SarQoL^®^ assess quality of life in people with sarcopenia, the impact of comorbidities and their interaction on functional capacity was not assessed, as it is a risk factor to consider and may influence the SarQoL^®^ results, so future adaptations and validations should take into account the inclusion of the different comorbidities that these subjects may present.

## 5. Conclusions

The main conclusion that can be drawn after carrying out the study is that the SarQoL^®^ has been translated and adapted into multiple languages. All the versions analyzed show basic psychometric characteristics that can be classified qualitatively ranging between good and excellent. However, only three of the 13 versions analyzed featured adequate methodological quality.

Clinicians and researchers at the international level have different instruments with psychometric properties that, as a rule, are similar to the adapted and validated versions existing of the SarQoL^®^ published in different languages. Therefore, these properties allow us to compare the results obtained with samples from different countries. Despite these good characteristics, there are psychometric variables that none of the versions included, affecting their rating criteria for good measurement properties and methodological quality according to the COSMIN guidelines. Therefore, it is necessary to design studies that include the same measurement properties so that the validation process is homogeneous within the scientific community. Likewise, thanks to this study, it can be concluded that the SarQoL^®^ questionnaires available so far, as tools for evaluating the quality of life in people with sarcopenia, which have been translated and validated in different languages, are valid to be used in this population in different countries.

## Figures and Tables

**Figure 1 ijerph-19-04561-f001:**
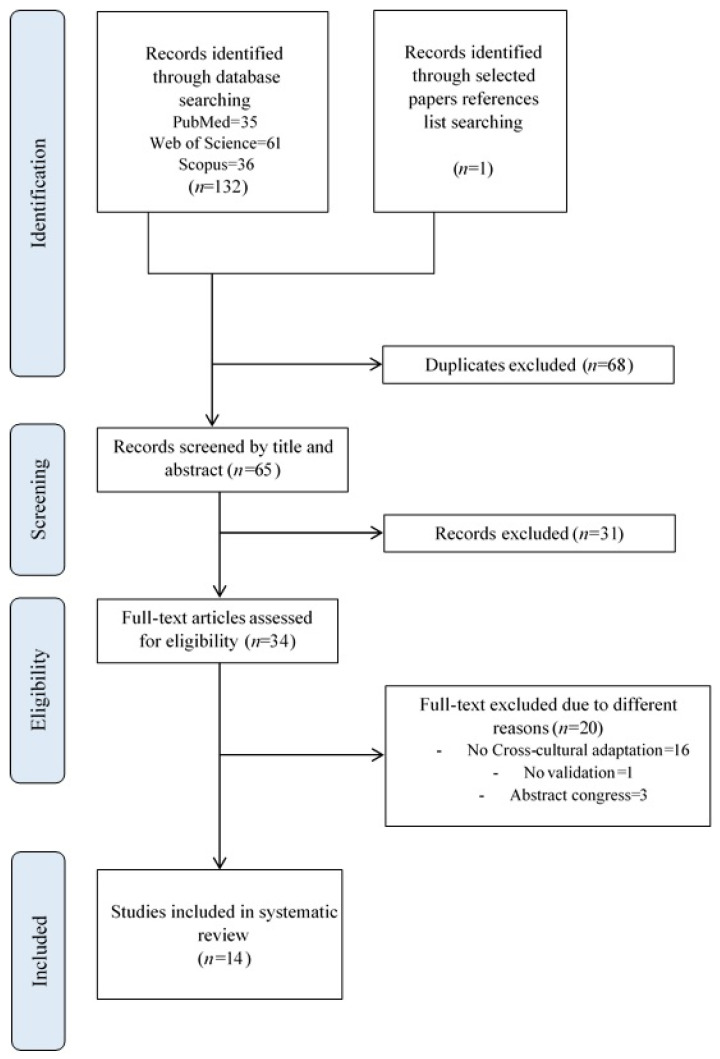
Flowchart for studies selection based on PRISMA.

**Table 1 ijerph-19-04561-t001:** Structural characteristics of the questionnaires.

Questionnaire/ Author, Year	Acronym	Population/BMI	Setting	Diagnosis of Sarcopenia/Number of Subjects with Sarcopenia	Number of Subjects—Phase Pilotage	Number of Subjects per Items
English translation and validation of the SarQoL^®^, a quality of life questionnaire specific for sarcopenia/Beaudart et al., 2017 [30]	NR	444 subjects (222 females, 222 males)/ 75.2 (2.6) years mean (SD)/ BMI (kg/m^2^): (28.1 (4.6) mean (SD)	Hertfordshire	EWGSOP/ Sarcopenia *n* = 14 *n* = 93 subjects with low “muscle function”.	10	8
Romanian Translation and Cross-Cultural Adaptation of the SarQol Questionnaire/Gasparik et al., 2016 [21]	NR	20 subjects (10 sarcopenic and 10 non sarcopenic with different educational and socioeconomic backgrounds)	Clinical County Hospital, Târgu Mureș	-	20	-
Psychometric performance of the Romanian version of the SarQoL^®^, a health-related quality of life questionnaire for sarcopenia/Gasparik et al., 2017 [31]	SarQoL^®^-Ro	100 subjects both sexes. Aged 65 years old or above/ BMI (kg/m^2^): <30	-	EWGSOP/ Sarcopenia *n* = 13 22.1 (19.8–23.1) Non-sarcopenia *n* = 87 26.6 (24.8–29.1)	20	2
Translation and validation of the Dutch SarQoL^®^, a quality of life questionnaire specific to sarcopenia/Geerinck et al., 2018 [22]	SarQoL^®^-NL	92 subjects (40 females and 52 males) 82(73–85) years/ BMI (kg/m^2^): 26.19 (23.05–29.00)	Gerontology Department of the Vrije Universiteit Brussel (VUB)	EWGSOP/ Sarcopenia *n* = 30 (13 females and 17 men)	14	<2
Polish Validation of the SarQoL^®^, a Quality of Life Questionnaire Specific to Sarcopenia/Konstantynowicz et al., 2018 [12]	SarQoL^®^-PL	106 subjects (65.1% females)/ Aged 73.3 (5.94) years Mean (SD) BMI (kg/m^2^): Sarcopenia 28.2 (4.92) Non-sarcopenia 29.7 (4.91)	Two outpatient clinics in Poland (Bialystok and Warsaw)	EWGSOP/ Sarcopenia: 60 subjects (43 females, 17 males)	10	2
Cross cultural adaptation of the Greek sarcopenia quality of life (SarQoL) questionnaire/Tsekoura et al., 2018 [23]	SarQoL ^GR^	176 Greek elderly people 136 females, 40 males aged 71.19 (7.95) years mean (SD)/ BMI (kg/m^2^): 26.6 (SD = 3.85)	The University Hospital of Rio, Greece, and the laboratory of Technological Educational Institute of Western Greece	EWGSOP/ Sarcopenia *n* = 50 (37 females, 13 males)	15	3
Validation of the Lithuanian version of sarcopenia-specific quality of life questionnaire (SarQoL^®^)/Alekna et al., 2019 [13]	NR	176 subjects (105 females, 71 males)/ Aged 78.2 (74.1–82.6) years/ BMI (kg/m^2^): 23.38 (21.91–25.22)	The National Osteoporosis Centre, an outpatient clinic in Vilnius, Lithuania	EWGSOP2/ Sarcopenia *n* = 58 (25 females, 33 males)	16	3
Russian translation and validation of SarQoL^®^ - quality of life questionnaire for patients with sarcopenia/Safonova et al., 2019 [24]	NR	100 subjects (70% females; 30% males) Aged 74.0 (6.5) years Mean (SD) BMI (kg/m^2^): NR	NR	EWGSOP/ Sarcopenia *n* = 50 (35 females, 15 males)	20	<2
Psychometric Properties of the Spanish Version of the Sarcopenia and Quality of Life, a Quality of Life Questionnaire Specific for Sarcopenia/Fábrega-Cuadros et al., 2020 [5]	NR	252 subjects (208 females, 44 males)/ Aged 74.00 (70.00–78.00) years/ BMI (kg/m^2^): NR	Two centers of active participation of older adults in Jaén, Spain	EWGSOP2/ Sarcopenia *n* = 66 (49 females, 17 males)	NR	<5
Cross-sectional Evaluation of the Sarcopenia Quality of Life (SarQoL) Questionnaire: Translation and Validation of its Psychometric Properties/Dzhus et al., 2020 [25]	SarQoL-UA	49 subjects (20 females, 29 males)/ Aged 71.00 (67.00–77.50) years/ BMI (kg/m^2^): 29.06 (25.28–32.62)	Oleksandrivska Clinical Hospital in Kyiv, Ukraine	EWGSOP2/ Probably sarcopenia *n* = 28 (12 females, 16 males)	10	<1
Translation and validation of the Korean version of the Sarcopenia Quality of Life (SarQoL-K^®^) questionnaire and applicability with the SARC-F screening tool/Yoo et al., 2020 [26]	SarQoL-K^®^	450 subjects (399 females, 51 males)/ Aged 73.9 (6.6) years mean (SD)/ BMI (kg/m^2^): NR	Six rural area	EWGSOP2/ Sarcopenia *n* = 53	10	8
Translation and psychometric performance of the Serbian version of the Sarcopenia Quality of Life (SarQoL^®^) questionnaire/Matijević et al., 2020 [27]	NR	699 subjects (508 females, 191 males)/ Aged 70 (67–74) years/ BMI (kg/m^2^): 29.41 (26.2–32.38)	Pensioners’ association of Novi Sad, Serbia	EWGSOP2/ Sarcopenia *n* = 12 (9 females, 3 males)	25	13
Psychometric Properties of the Chinese Version of the Sarcopenia and Quality of Life, a Quality of Life Questionnaire Specific for Sarcopenia/Le et al., 2021 [28]	SarQoL^®^-CN	159 subjects (74 females, 85 males)/ Aged sarcopenia: 80.16 (7.42); Aged non-sarcopenia: 70.00 (66.00–74.75)/ BMI (kg/m^2^): sarcopenia 19.08 (2.16) mean (SD)	Honghui hospital, Xi’an Jiaotong University, China	AWGS 2019 consensus/Sarcopenia *n* = 51 (39 females)	10	<3
Sarcopenia quality-of-life questionnaire (SarQoL)^®^: translation, cross-cultural adaptation and validation in Turkish/Erdogan et al., 2021 [29]	SarQoL^®^-TR	100 subjects (71 females, 29 males)/ Aged: 74.7 (6.1) years/ BMI (kg/m^2^): 28.7 (5.4) mean (SD)	Geriatric outpatient clinics at two different university hospitals	EWGSOP2/Probable sarcopenia *n* = 27; Confirmed sarcopenia *n* = 5; Severe sarcopenia *n* = 4	10	<2

BMI: Body mass index; NR: Not Reported; EWGSOP: European Working Group on Sarcopenia in Older People; EWGSOP2: European Working Group on Sarcopenia in Older People, and revised in early 2018; AWGS: Asian Working Group for Sarcopenia criteria.

**Table 2 ijerph-19-04561-t002:** Psychometric properties of the questionnaires.

Study/Version	Test-Retest Reliability	Internal Consistency	Construct Validity	
			Convergent Validity r (*p*-Value)	Divergent Validity r (*p*-Value)/or IC Range
Beaudart et al., 2017 [30]/English version	ICC = 0.95 (95% CI 0.92–0.97) D6 ICC = 0.78 (0.58–0.88) lowest domain score	Cronbach’s α = 0.88	SF-36 physical functioning 0.82 (<0.001) SF-36 role limitation due to physical problem 0.54 (<0.001) SF-36 bodily pain 0.55 (<0.001) SF-36 general health 0.49 (<0.001) SF-36 vitality 0.74 (<0.001) EQ-5D utility score 0.58 (<0.001) EQ-5D mobility −0.56 (<0.001) EQ-5D usual activities −0.55 (<0.001)	SF-36 social functioning 0.47 (0.001) SF-36 role limitation due to emotional problem 0.22 (0.04) SF-36 mental health 0.29 (0.007) EQ-5D self-care −0.24 (0.032) EQ-5D pain/discomfort −0.41 (<0.001) EQ-5D anxiety/depression −0.32 (0.004)
Gasparik et al., 2017 [31]/Romanian version	NR	Cronbach’s α = 0.946	SF-36 physical functioning 0.8903 (<0.0001) SF-36 role limitations due to physical health 0.6763 (<0.0001) SF-36 pain 0.5715 (0.0006) SF-36 general health 0.6943 (<0.0001) SF-36 vitality 0.8951 (<0.0001) EQ-5D usual activities −0.6106 (0.0002) EQ-5D mobility −0.6893 (<0.0001)	SF-36 social functioning 0.5765 (0.0006) SF-36 Role limitations due to emotional problems 0.5031 (0.0033) SF-36 mental health 0.6822 (0.0001) EQ-5D self-care −0.5356 (0.0016) EQ-5D anxiety/depression −0.4240 (0.0156) EQ-5D pain/discomfort −0.4580 (0.0084)
Geerinck et al., 2018 [22]/Dutch version	ICC = 0.976 (95% CI 0.947–0.989) D1 ICC = 0.820 (0.642–0.915) D2 ICC = 0.98 (0.793–0.959) D3 ICC = 0.707 (0.447–0.857) D4 ICC =0.948 (0.888–0.976) D5 ICC = 0.875 (0.741–0.942) D6 ICC = 0.375 (0.001–0.660) D7 ICC = 0.235 (−0.617–0.568)	Cronbach’s α = 0.883	SF-36 physical functioning 0.842 (<0.001) SF-36 role limitation physical 0.551 (0.002) SF-36 body pain 0.546 (0.002) SF-36 general health 0.617 (<0.001) SF-36 vitality 0.647 (<0.001) EQ-5D utility score 0.771 (<0.001) EQ-5D mobility −0.749 (<0.001) EQ-5D usual activities −0.575 (0.001) EQ-VAS 0.780 (<0.001)	SF-36 Social functioning 0.426 (0.019) SF-36 Role limitation emotional 0.594 (0.001) SF-36 mental health 0.430 (0.018) EQ-5D self-care −0.520 (0.003) EQ-5D pain/discomfort −0.418 (0.024) EQ-5D anxiety/depression −0.225 (0.223)
Konstantynowicz et al., 2018 [12]/Polish version	ICC = 0.99 (95% CI 0.995–0.999) D1 ICC = 0.98 (0.96–0.99) D2 ICC = 0.99 (0.990–0.997) D3 ICC = 0.98 (0.97–0.99) D4 ICC = 0.99 (0.986–0.996) D5 ICC = 0.98 (0.96–0.99) D6 ICC = 1.00 D7 ICC = 0.96 (0.92–0.98)	Cronbach’s α = 0.92	SF-36 v2 PCS 0.88 (<0.001) SF-36 v2 MCS 0.62 (<0.001) EQ-5D index value 0.72 (<0.001) EQ-VAS 0.71 (<0.001)	NR
Tsekoura et al., 2018 [23]/Greek version	ICC = 0.96 (95% CI 0.95–0.97) D1 ICC = 0.97 (0.97–0.98) D2 ICC = 0.98 (0.97–0.98) D3 ICC = 0.84 (0.79–0.88) D4 ICC = 0.97 (0.88–0.98) D5 ICC = 0.91 (0.88–0.93) D6 ICC = 0.91 (0.89–0.94) D7 ICC = 0.64 (0.52–0.70)	Cronbach’s α = 0.96	SF-36 physical functioning 0.9 (<0.001) SF-36 bodily pain 0.53 (<0.001) SF-36 general health 0.42 (<0.001) SF-36 vitality 0.45 (<0.001) EQ-5D utility score 0.77 (<0.001) EQ-5D mobility 0.48 (<0.001) EQ-5D usual activities 0.62 (<0.001)	SF-36 social functioning 0.27 (0.02–0.53) SF-36 mental health 0.88 (0.75–0.94) SF-36 role limitation due to physical problems 0.41 (0.18–0.63) SF-36 role limitation due to emotional problems 0.33 (0.12–0.98) EQ-5D pain/discomfort0.46 (0.10–0.74) EQ-5D anxiety/depression 0.55 (0.32–0.77) EQ- 5D self-care 0.44 (0.23–0.60)
Alekna et al, 2019 [13]/Lithuanian version	ICC = 0.976 (95% CI 0.959–0.986) D1 ICC = 0.939 (0.898–0.964) D2 ICC = 0.957 (0.927–0.975) D3 ICC = 0.956 (0.925–0.973) D4 ICC = 0.969 (0.947–0.982) D5 ICC = 0.987 (0.978–0.993) D6 ICC = 0.854 (0.761–0.913) D7 ICC = 0.875 (0.793–0.926)	Cronbach’s α = 0.95 D1 Cronbach’s α = 0.94 D2 Cronbach’s α = 0.94 D3 Cronbach’s α = 0.95 D4 Cronbach’s α = 0.94 D5 Cronbach’s α = 0.95 D6 Cronbach’s α = 0.96 D7 Cronbach’s α = 0.95	SF-36 physical functioning 0.554 (<0.001) SF-36 role limitation due to physical problems 0.519 (<0.001) SF-36 vitality 0.559 (<0.001) EQ-5D utility score 0.576 (<0.001)	SF-36 role limitation due to emotional problems 0.362 (0.001) SF-36 mental health 0.364 (0.005) EQ-5D self-care −0.391 (<0.001) EQ-5D anxiety/depression −0.369 (<0.001)
Safonova et al., 2019 [24]/Russian version	ICC = 0.935 (95% CI 0.91–0.96) D6 ICC = 0.73 (0.58–0.88) (Lower index)	Cronbach’s α = 0.924	SF-36 physical functioning 0.63 (<0.001) SF-36 role limitation due to physical problems 0.39 (0.0046) SF-36 body pain 0.27 (0.06) SF-36 general health 0.40 (0.0045) SF-36 vitality 0.29 (0.042) EQ-5D utility score 0.53 (<0.0001) EQ-5D mobility 0.53 (<0.0001) EQ-5D usual activities 0.54 (<0.0001)	SF-36 social functioning 0.34 (0.017) SF-36 role limitation due to emotional problems 0.23 (0.10) SF-36 mental health 0.07 (0.62) EQ-5D self-care 0.53 (<0.0001) EQ-5D pain/discomfort 0.52 (<0.0001) EQ-5D anxiety/depression 0.53 (<0.0001)
Fábrega-Cuadros et al., 2020 [5]/Spanish version	ICC = 0.99 (95% CI 0.98–0.99) D1 ICC = 0.98 (0.96–0.99) D2 ICC = 0.99 (0.98–0.99) D3 ICC = 0.99 (0.98–1.00) D4 ICC = 0.98 (0.97–0.99) D5 ICC = 0.96 (0.92–0.98) D6 ICC = 0.88 (0.80–0.93) D7 ICC = 0.84 (0.72–0.91)	Cronbach’s α = 0.904	SF-36 physical functioning 0.53 (<0.001) SF-36 role limitation due to physical problems 0.38 (0.002) SF-36 general health 0.42 (<0.001) SF-36 body pain 0.36 (0.003) SF-36 vitality 0.50 (<0.001) EQ-5D-3L mobility −0.50 (<0.001) EQ-5D-3L usual activities −0.40 (0.001) EQ-5D-3L VAS 0.49 (<0.001) EQ-5D-3L utility score r = 0.41	HADS anxiety −0.11 (−0.35 to 0.14) (*p* = 0.38) HADS depression −0.18 (0.39 to 0.05) (*p* = 0.149) EQ-5D-3L self-care −0.16 (−0.34 to 0.08) (*p* = 0.199) EQ-5D-3L pain/discomfort −0.17 (−0.40 to 0.10) (*p* = 0.162) EQ-5D-3L anxiety/depression −0,31 (−0,51 to 0,36) (*p* = 0.013)
Dzhus et al., 2020 [25]/Ukrainian version	ICC = 0.997 (95% CI 0.994–0.998) D1 ICC = 0.992 (0.985–0.995) D2 ICC = 0.995 (0.990–0.997) D3 ICC = 0.990 (0.982–0.994) D4 ICC = 0.986 (0.976–0.992) D5 ICC = 0.995 (0.991–0.997) D6 ICC = 0.950 (0.913–0.971) D7 ICC = 0.933 (0.884–0.961)	Cronbach’s α = 0.898 D1 Cronbach’s α = 0.872 D2 Cronbach’s α = 0.874 D3 Cronbach’s α = 0.874 D4 Cronbach’s α = 0.861 D5 Cronbach’s α = 0.875 D6 Cronbach’s α = 0.912 D7 Cronbach’s α = 0.901	SF-36 PCS 0.833 (<0.001) EQ-5D mobility −0.794 (<0.001) EQ-5D usual activities −0.677 (0.001) EQ-5D VAS 0.466 (0.001)	Complete sample (*n* = 49) SF-36 MCS 0.295 (0.039) EQ-5D self-care −0.632 (<0.001) EQ-5D pain/discomfort −0.650 (<0.001) EQ-5D anxiety/depression −0.454 (0.001) Probably sarcopenic sample (*n* = 28) SF-36 MCS 0.177 (0.367) EQ-5D self-care −0.700 (<0.001) EQ-5D pain/discomfort −0.684 *p* < 0.001 EQ-5D anxiety/depression −0.423 *p* = 0.025
Yoo et al., 2020 [26]/ Korean version	ICC = 0.977 (95% CI 0.975–0.979) D1 ICC = 0.966 (0.950–0.980) D2 ICC = 0.993 (0.990–0.997) D3 ICC = 0.981 (0.970–0.990) D4 ICC = 0.991 (0.986–0.996) D5 ICC = 0.981 (0.960–0.990) D6 ICC = 0.860 (0.740–0.930) D7 ICC = 0.960 (0.920–0.980)	Cronbach’s α = 0.866	SF-36 physical functioning 0.807 (<0.0001) SF-36 vitality 0.326 (<0.0001) SF-36 body pain 0.724 (<0.0001) SF-36 general health 0.607 (<0.0001) SF-36 role limitation due to physical 0.765 (<0.0001) EQ-5D utility score 0.468 (<0.0001) EQ-5D mobility −0.446 (<0.0001) EQ-5D usual activities −0.429 (<0.0001)	SF-36 emotional wellbeing −0.058 (−0.150–0.034) (*p* =0.217) EQ-5D self-care −0.120 (−0.200–0.012) (*p* = 0.231) EQ-5D pain/discomfort −0.287 (−0.355- -0213) (*p* = 0.045) EQ- 5D anxiety/depression −0.072 (−0.149–0.016) (*p* =0.478)
Matijević et al., 2020 [27]/Serbian version	NR	Cronbach’s α = 0.87	SF-36 physical functioning 0.760 (0.002) SF-36 role limitation due to physical 0.637 (0.001) SF-36 vitality 0.656 (0.005) EQ-5D index score 0.589 (<0.001)	SF-36 role limitation due to emotional problems 0.490 (<0.001) SF-36 mental health 0.474 (<0.001) EQ-5D anxiety -0.332 (<0.001) EQ-5D self-care -0.332 *p* < 0.001
Le et al., 2021 [28]/ Chinese version	ICC = 0.936 (95% CI (0.994–0.998) D1 ICC = 0.985 (0.974–0.991) D2 ICC = 0.996 (0.994–0.998) D3 ICC = 0.968 (0.945–0.981) D4 ICC = 0.997 (0.995–0.998) D5 ICC = 0.987 (0.978–0.988) D6 ICC = 1 D7 ICC = 0.936 (0.891–0.963)	Cronbach’s α = 0.867	SF-36 physical functioning 0.824 (<0.001) SF-36 role limitation due to physical health 0.756 (<0.001) SF-36 bodily pain 0.250 (0.077) SF-36 general health 0.557 (<0.001) SF-36 vitality 0.401 (0.004) EQ-5D mobility −0.804 (<0.001) EQ-5D usual activities −0.864 (<0.001)	SF-36 social functioning 0.725 (<0.001) SF-36 role limitations due to emotional problems 0.440 (0.001) SF-36 mental health 0.344 (0.014) EQ-5D self-care −0.823 (<0.001) EQ-5D pain/discomfort −0.114 (0.425) EQ-5D anxiety/depression −0.421 (0.002)
Erdogan et al., 2021 [29]/Turkish version	ICC = 0.97 (95% CI 0.94–0.98) D1 ICC = 0.89 (0.81–0.94) D2 ICC = 0.96 (0.92–0.98) D3 ICC = 0.88 (0.78–0.93) D4 ICC = 0.96 (0.93–0.98) D5 ICC = 0.97 (0.95–0.99) D6 ICC = 0.85 (0.72–0.92) D7 ICC = 0.85 (0.72–0.92)	Cronbach’s α = 0.88	SF-36 physical functioning 0.82 (<0.001) SF-36 role limitation due to physical problems 0.69 (<0.001) SF-36 general health 0.60 (<0.001) SF-36 vitality 0.69 (<0.001) EQ-5D mobility −0.59 (<0.001) EQ-5D selfcare −0.59 (<0.001) EQ-5D usual activities −0.63 (<0.001)	SF-36 social functioning 0.50 (<0.001) SF-36 role of limitation due to emotional problems 0.50 (<0.001) SF-36 mental health 0.56 (<0.001) SF-36 bodily pain 0.48 (<0.001) EQ-5D pain/discomfort −0.56 (<0.001) EQ-5D anxiety/depression −0.45 (<0.001)

NR: not reported.

**Table 3 ijerph-19-04561-t003:** Rating of the psychometric properties and methodological quality.

From	Country (Language) in Which the Questionnaire Was Valuated	Measurement Error	Internal Consistency	Hypotheses Testing		Reliability	Methodological Quality
		Rating	Rating	Rating Convergent	Rating Divergent	Rating	
Beaudart et al., 2017 [30]	English	NA	Indeterminate	Sufficient	Insufficient	Sufficient	Very good
Gasparik et al., 2017 [31]	Romanian	Indeterminate	Indeterminate	Sufficient	Insufficient	Sufficient	Inadequate
Geerinck et al., 2018 [22]	Dutch	NA	Indeterminate	Sufficient	Insufficient	Sufficient	Doubtful
Konstantynowicz et al., 2018 [12]	Polish	NA	Indeterminate	Sufficient	Indeterminate	Sufficient	Inadequate
Tsekoura et al., 2018 [23]	Greek	Indeterminate	Indeterminate	Insufficient	Insufficient	Sufficient	Inadequate
Alekna et al., 2019 [13]	Lithuanian	NA	Indeterminate	Sufficient	Insufficient	Sufficient	Inadequate
Safonova et al., 2019 [24]	Russian	NA	Indeterminate	Insufficient	Insufficient	Sufficient	Inadequate
Fábrega-Cuadros et al., 2020 [5]	Spanish	NA	Indeterminate	Insufficient	Sufficient	Sufficient	Inadequate
Dzhus et al., 2020 [25]	Ukrainian	NA	Indeterminate	Sufficient	Insufficient	Sufficient	Inadequate
Yoo et al., 2020 [26]	Korean	NA	Indeterminate	Insufficient	Sufficient	Sufficient	Very good
Matijević et al., 2020 [27]	Serbian	NA	Indeterminate	Sufficient	Insufficient	Sufficient	Very good
Le et al., 2021 [28]	Chinese	NA	Indeterminate	Sufficient	Insufficient	Sufficient	Inadequate
Erdogan et al., 2021 [29]	Turkish	NA	Indeterminate	Sufficient	Insufficient	Sufficient	Inadequate

NA: Not applicable.

## Data Availability

Data are available upon request.

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
