# Peer review of "Analysis of Structural Characteristics and Psychometric Properties of the SarQoL® Questionnaire in Different Languages: A Systematic Review"

_ijerph, 2022, doi:10.3390/ijerph19084561_

Round 1

Reviewer 1 Report

This is a well-written systemic review of the literature. The authors attempt to synthesize current knowledge on the SarQoL questionnaires developed in different languages.

The synthesis of knowledge is clear and comprehensive. This work is essential in light of the need to improve our understanding of the impact of sarcopenia on QoL of older adults.

Comments:

Table 1: For the population characteristics of each study, I would recommend adding the following information: setting (e.g., community or hospital, etc.); diseases (e.g., the average number of diseases each participant had).

Author Response

RESPONSE TO REVIEWERS: Itemized List

INTERNATIONAL JOURNAL OF ENVIRONMENTAL RESEARCH AND PUBLIC HEALTH

SPECIAL ISSUE: Age-Related Sarcopenia, Obesity and Inflammaging: Effects of Physical Activity and Nutrition

Manuscript ID IJERPH-1631450

We would like to thank the Editor and reviewers for their thoughtful and constructive comments. We have considered all suggestions, and have incorporated them into the revised manuscript. Changes to the original manuscript are identified by highlights (in yellow background). After corrections made, we believe that our document is much easier to read and understand. An itemized point-by-point response to the reviewers’ comments is presented below. 

Thank you very much for offering us the possibility of reviewing the document and being able to complement it with the suggestions and comments made by the reviewers. We have followed all the suggestions made by the reviewer to understand that the document evolves positively.

Reviewer: 1

Table 1: For the population characteristics of each study, I would recommend adding the following information: setting (e.g., community or hospital, etc.); diseases (e.g., the average number of diseases each participant had).

Authors’ answer: Thank you for your suggestion. This has been modified within the table 1. A setting column has been included. On the other hand, we appreciate the suggestion to include diseases but the authors have considered not including them because most of the studies include them in the exclusion criteria.

Reviewer 2 Report

Thank you so much for inviting me to review this manuscript. Authors tried to examine and explain the topics of the SarQoL® in different languages in people with sarcopenia and to review their psychometric properties. Congratulations to the authors for their effort and work. I have some comments to improve the manuscript:

- I consider that the importance of this work needs to be argued further. The introduction is a bit brief.

- I understood that the authors only selected studies written in English. "Documents published in languages other than English were excluded." If the objective was: "The main objective of this review was to examine and explain the topics of the SarQoL in different languages in people with sarcopenia and to review their psychometric properties". Why did the authors only consider those in English? Any justification? I think the paper would benefit from including a wider variety of languages (to provide more accurate information).

- The limitations should be expanded, as well as highlighting the possible strengths of the study. Again, what is new about this study in the scientific literature?

- “The main conclusion that can be drawn from this study is that the different versions of the questionnaires show good structural characteristics and basic psychometric properties for the evaluation of the quality of life in patients with sarcopenia”. The authors have not taken into account that only 3 of the 13 studies included in the systematic review were of good methodological quality. This seems to me to be too ambitious a conclusion in relation to this fact. Caution is required here.

Best wishes,

Author Response

RESPONSE TO REVIEWERS: Itemized List

INTERNATIONAL JOURNAL OF ENVIRONMENTAL RESEARCH AND PUBLIC HEALTH

SPECIAL ISSUE: Age-Related Sarcopenia, Obesity and Inflammaging: Effects of Physical Activity and Nutrition

Manuscript ID IJERPH-1631450

We would like to thank the Editor and reviewers for their thoughtful and constructive comments. We have considered all suggestions, and have incorporated them into the revised manuscript. Changes to the original manuscript are identified by highlights (in yellow background). After corrections made, we believe that our document is much easier to read and understand. An itemized point-by-point response to the reviewers’ comments is presented below. 

Thank you very much for offering us the possibility of reviewing the document and being able to complement it with the suggestions and comments made by the reviewers. We have followed all the suggestions made by the reviewer to understand that the document evolves positively.

Reviewer: 2

- I consider that the importance of this work needs to be argued further. The introduction is a bit brief.

Authors’ answer: Thank you very much for your appreciation. The introduction has been modified and relevant sections have been added to better justify the development of this revision.

- I understood that the authors only selected studies written in English. "Documents published in languages other than English were excluded." If the objective was: "The main objective of this review was to examine and explain the topics of the SarQoL in different languages in people with sarcopenia and to review their psychometric properties". Why did the authors only consider those in English? Any justification? I think the paper would benefit from including a wider variety of languages (to provide more accurate information).

Authors’ answer:  Thank you for your suggestion. Regarding the Hungarian version, the authors decided to exclude it since the validation phase was not performed, only the cross-cultural adaptation was developed. Following the reviewer's recommendations, the Russian version has been included despite not being written in English because it enriches this review, and we appreciate the appreciation. The Russian version has been included in the tables and throughout the manuscript.

- The limitations should be expanded, as well as highlighting the possible strengths of the study. Again, what is new about this study in the scientific literature?

Authors’ answer: Thank you very much for your questions. We have renamed the section by strengths and weaknesses. We have added the main strength of the study and some weaknesses that need to be taken into account when interpreting the results presented in this study.  

- “The main conclusion that can be drawn from this study is that the different versions of the questionnaires show good structural characteristics and basic psychometric properties for the evaluation of the quality of life in patients with sarcopenia”. The authors have not taken into account that only 3 of the 13 studies included in the systematic review were of good methodological quality. This seems to me to be too ambitious a conclusion in relation to this fact. Caution is required here.

Authors’ answer: Thank you very much for the suggestion. We agree with the reviewer that this aspect should be included in the conclusion. For this reason, we have partially rewritten the conclusion.

Reviewer 3 Report

In systematic review the authors investigate about the characteristics and psychometric and measurement properties of the SarQoL®, its methodological quality and the quality of the evidence of the adaptations and validations of the questionnaire in different languages.

Abstract

From “of heterogeneous” to “population” rewrite more clearly.

Write “Web of science” instead of WOS.

You write “Methodological quality was assessed based on the COSMIN scale”. Are you talking about the methodological quality of the studies included in your review? The Cosmin is not a tool that evaluate the general quality of studies; how did you asses wheater if the studies that you include in your review are worth it or not? Please explain this bias.

The aim of your study cited in the abstract is not congruent with the aim mentioned in the end of your introduction section. You also write:” The main objective of this review was to examine and explain the topics of the SarQoL® in different languages in people with sarcopenia and to review their psychometric properties”. What does it mean that the aim of your study is to “explain the topics of the SarQol®”?  Rewrite the aim more clearly.

Introduction

The introduction provided a good insight on sarcopenia but it does not talk enough about the SarQol®, provide more information on it. The overall issue of the introduction is that many periods are not wrote correctly and, due to this, the information are difficult to understand. I suggest an important revision of the English.

Methods

In “sources and search” subsection specify better the terms of your search strategy.

In “selection criteria” give a deeper insight on your inclusion and exclusion criteria.

The risk of bias assessment is lacking, please add it.

Results

You mentioned that a document “very relevant” for your review was not indexed in any of the three major databases (e.g. Pubmed, Scopus, Web of Science) but you did not specify which document is. Please provide the reference to this document. Moreover, the sentence: “(document that is not

indexed in any of the three selected databases, but that is very relevant for this review)” is not needed.

In table 1 there are no translated words (e.g. mujeres, hombres) please fix it.

Table 2 is written in italics, fix it.

Under table 1 you repeat too much the “on the other hand”. Write this subparagraph more academically.

Discussion

The content of this section is appropriate, but a revision of the English language is required to give a better understanding to the reader.

Conclusion

Conclusion needs to be more concise.

Author Response

RESPONSE TO REVIEWERS: Itemized List

INTERNATIONAL JOURNAL OF ENVIRONMENTAL RESEARCH AND PUBLIC HEALTH

SPECIAL ISSUE: Age-Related Sarcopenia, Obesity and Inflammaging: Effects of Physical Activity and Nutrition

Manuscript ID IJERPH-1631450

We would like to thank the Editor and reviewers for their thoughtful and constructive comments. We have considered all suggestions, and have incorporated them into the revised manuscript. Changes to the original manuscript are identified by highlights (in yellow background). After corrections made, we believe that our document is much easier to read and understand. An itemized point-by-point response to the reviewers’ comments is presented below. 

Thank you very much for offering us the possibility of reviewing the document and being able to complement it with the suggestions and comments made by the reviewers. We have followed all the suggestions made by the reviewer to understand that the document evolves positively.

Reviewer: 3

Abstract

From “of heterogeneous” to “population” rewrite more clearly.

Authors’ answer:  Thank you very much for your appreciation, the wording of the sentence has been improved for better understanding.

Write “Web of science” instead of WOS.

Authors’ answer:  Thank you for your suggestion. This has been modified within the abstract.

You write “Methodological quality was assessed based on the COSMIN scale”. Are you talking about the methodological quality of the studies included in your review? The Cosmin is not a tool that evaluate the general quality of studies; how did you asses wheater if the studies that you include in your review are worth it or not? Please explain this bias.

Authors’ answer:  Thank you very much for your appreciation. When the authors refer to methodological quality, it is in relation to the structural characteristic included in table 1 (and reflected in table 3 as methodological quality), where reference is made to the number of patients per item included in the COSMIN guidelines.

The aim of your study cited in the abstract is not congruent with the aim mentioned in the end of your introduction section. You also write:” The main objective of this review was to examine and explain the topics of the SarQoL® in different languages in people with sarcopenia and to review their psychometric properties”. What does it mean that the aim of your study is to “explain the topics of the SarQol®”?  Rewrite the aim more clearly.

Authors’ answer: Thank you for your suggestion. The objective of the abstract has been modified and expressed more clearly and congruently with the objective that appears in the manuscript.

Introduction

The introduction provided a good insight on sarcopenia but it does not talk enough about the SarQol®, provide more information on it. The overall issue of the introduction is that many periods are not wrote correctly and, due to this, the information are difficult to understand. I suggest an important revision of the English.

Authors’ answer:  Thank you very much for your appreciation. The introduction has been modified and relevant sections have been added to better justify the development of this revision. We apologize for the English editing; it is not the native language of any of the authors. Again, we have had the collaboration of a professional Spanish-English translator for the revision of the English edition.

Methods

In “sources and search” subsection specify better the terms of your search strategy.

Authors’ answer: Thank you for your suggestion. We have included the terms that we have used to better define the search strategy.

In “selection criteria” give a deeper insight on your inclusion and exclusion criteria.

Authors’ answer: Thank you for your suggestion. This section has been modified to provide more detailed selection criteria.

The risk of bias assessment is lacking, please add it.

Authors’ answer: Thank you for your suggestion. We have included in the section of strengths and weaknesses some aspects / biases, which must be taken into account when interpreting the results.

Results

You mentioned that a document “very relevant” for your review was not indexed in any of the three major databases (e.g. Pubmed, Scopus, Web of Science) but you did not specify which document is. Please provide the reference to this document. Moreover, the sentence: “(document that is not indexed in any of the three selected databases, but that is very relevant for this review)” is not needed.

Authors’ answer: Thank you for your suggestion. The sentence has been deleted and the reference of the selected document has been specified.

In table 1 there are no translated words (e.g. mujeres, hombres) please fix it.

Authors’ answer: Thank you very much for your appreciation and sorry for the translation error. It has been modified in table 1.

Table 2 is written in italics, fix it.

Authors’ answer: Thank you very much for your appreciation, we regret the typographical error. It has been modified in table 2.

Under table 1 you repeat too much the “on the other hand”. Write this subparagraph more academically.

Authors’ answer: Thank you very much for your appreciation. The term has been modified.

Discussion

The content of this section is appropriate, but a revision of the English language is required to give a better understanding to the reader.

Authors’ answer:  We apologize for the English edition. We have resent the manuscript to a professional Spanish-English translator for correction.

Conclusion

Conclusion needs to be more concise.

Authors’ answer: Thank you for your suggestion. This has been modified within the text to clarify this point.

Reviewer 4 Report

I am reviewing the article “Analysis of Structural Characteristics and Psychometric Properties of the SarQoL® Questionnaire in Different Languages: A Systematic Review”. This Systematic Review is an interesting article on an important topic in International Journal of Environmental Research and Public Health. However, there are a few concerns.

Introduction
1. The argument in the introduction is weak. It does not delve into the meaning and use of the SarQoL® concept. Also, they do not make an exposition of what is the current state of the subject.

 Selection criteria
2. Why wasn't the full text available? Please clarify.

Discussion
3. Much of the information that has been included in the discussion should have been covered in the introduction.

Author Response

RESPONSE TO REVIEWERS: Itemized List

INTERNATIONAL JOURNAL OF ENVIRONMENTAL RESEARCH AND PUBLIC HEALTH

SPECIAL ISSUE: Age-Related Sarcopenia, Obesity and Inflammaging: Effects of Physical Activity and Nutrition

Manuscript ID IJERPH-1631450

We would like to thank the Editor and reviewers for their thoughtful and constructive comments. We have considered all suggestions, and have incorporated them into the revised manuscript. Changes to the original manuscript are identified by highlights (in yellow background). After corrections made, we believe that our document is much easier to read and understand. An itemized point-by-point response to the reviewers’ comments is presented below. 

Thank you very much for offering us the possibility of reviewing the document and being able to complement it with the suggestions and comments made by the reviewers. We have followed all the suggestions made by the reviewer to understand that the document evolves positively.

Reviewer: 4

Introduction
1. The argument in the introduction is weak. It does not delve into the meaning and use of the SarQoL® concept. Also, they do not make an exposition of what is the current state of the subject.

Authors’ answer:  Thank you very much for your appreciation. The introduction has been modified and relevant sections have been added to better justify the development of this revision.

Selection criteria
2. Why wasn't the full text available? Please clarify.

Authors’ answer:  We appreciate the appreciation but the full text wasn´t available because it was only published as a conference abstract. The full text hasn´t been found in any database.

Discussion
3. Much of the information that has been included in the discussion should have been covered in the introduction.

Authors’ answer:  Thank you very much for your appreciation. Taking into account your suggestion, the authors have included in the introduction the information related to the development of the discussion.

Round 2

Reviewer 1 Report

The authors have addressed some concerns in this version of the manuscript.

Comments:

  1. Discussion: According to the authors' literature review, most studies excluded older adults with multiple chronic diseases. In other words, there is limited information on the questionnaire's psychometric properties in older adults with various chronic diseases, according to the authors' literature review. Without further studies addressing this issue in the future, the usefulness of the questionnaire in this sizable population of older adults would be questionable. Therefore, a discussion of this issue is suggested.

Author Response

RESPONSE TO REVIEWERS: Itemized List

INTERNATIONAL JOURNAL OF ENVIRONMENTAL RESEARCH AND PUBLIC HEALTH

SPECIAL ISSUE: Age-Related Sarcopenia, Obesity and Inflammaging: Effects of Physical Activity and Nutrition

Manuscript ID IJERPH-1631450

We would like to thank the Editor and reviewers for their thoughtful and constructive comments. We have considered all suggestions, and have incorporated them into the revised manuscript. Changes to the original manuscript are identified by highlights (in green background). After corrections made, we believe that our document is much easier to read and understand. An itemized point-by-point response to the reviewers’ comments is presented below. 

Thank you very much for offering us the possibility of reviewing the document and being able to complement it with the suggestions and comments made by the reviewers. We have followed all the suggestions made by the reviewer to understand that the document evolves positively.

Reviewer: 1

Discussion: According to the authors' literature review, most studies excluded older adults with multiple chronic diseases. In other words, there is limited information on the questionnaire's psychometric properties in older adults with various chronic diseases, according to the authors' literature review. Without further studies addressing this issue in the future, the usefulness of the questionnaire in this sizable population of older adults would be questionable. Therefore, a discussion of this issue is suggested.

Authors’ answer: Thank you for your suggestion. We have taken their assessment into account and it has been included in the discussion section as a limitation of the study and to be taken into account for future adaptations and validations of the SarQoL. 

Reviewer 2 Report

Authors have addresses all my comments. This manuscript can be accepted for publication.

Kind regards,

Author Response

Dear Reviewer, 

Thank you very much for your suggestions and contributions, they have greatly enriched the manuscript. 

Best regards.

Reviewer 3 Report

The authors correctly addressed the previous comments and provided an interesting version of the paper. 

Best wishes,

Author Response

(The authors gave the same response as above.)
